# Intergenerational transmission of polygenic predisposition for neuropsychiatric traits on emotional and behavioural difficulties in childhood

A. G. Allegrini [1,2] ✉, L. J. Hannigan [3,4,5], L. Frach [1,6], W. Barkhuizen [1], J. R. Baldwin [1,2], O. A. Andreassen [7], D. Bragantini[3,4], L. Hegemann[3,4,8], A. Havdahl[3,4,9,10] & J-B. Pingault [1,2,10]

Childhood emotional and behavioural difficulties tend to co-occur and often precede diagnosed neuropsychiatric conditions. Identifying shared and specific risk factors for early-life mental health difficulties is therefore essential for prevention strategies. Here, we examine how parental risk factors shape their offspring's emotional and behavioural symptoms (e.g. feelings of anxiety, and restlessness) using data from 14,959 genotyped family trios from the Norwegian Mother, Father and Child Cohort Study (MoBa). We model maternal reports of emotional and behavioural symptoms, organizing them into general and specific domains. We then investigate the direct (genetically transmitted) and indirect (environmentally mediated) contributions of parental polygenic risk for neuropsychiatric-related traits and whether these are shared across symptoms. We observe evidence consistent with an environmental route to general symptomatology beyond genetic transmission, while also demonstrating domain-specific direct and indirect genetic contributions. These findings improve our understanding of early risk pathways that can be targeted in preventive interventions aiming to interrupt the intergenerational cycle of risk transmission.

Emotional and behavioural difficulties onset during childhood or adolescence for most individuals[1]. In turn, symptoms of emotional and behavioural difficulties that manifest in early life may be harbingers of later-onset psychiatric disorders. An aetiological understanding of emotional and behavioural difficulties is thus central to advancing our understanding of how neuropsychiatric conditions develop.

Parental factors, particularly parental neuropsychiatric conditions, are considered important early risk factors contributing to childhood neuropsychiatric symptomatology[2]. The transmission of

[1]Division of Psychology and Language Sciences, Department of Clinical, Educational and Health Psychology, University College London, London, UK. [2]Social, Genetic and Developmental Psychiatry Centre, Institute of Psychiatry, Psychology and Neuroscience, King's College London, London, UK. [3]Research Department, Lovisenberg Diaconal Hospital, Oslo, Norway. [4]PsychGen Center for Genetic Epidemiology and Mental Health, Norwegian Institute of Public Health, Oslo, Norway. [5]MRC Integrative Epidemiology Unit (IEU), University of Bristol, Bristol, United Kingdom. [6]Institute of Human Genetics, University of Bonn, School of Medicine & University Hospital Bonn, Bonn, Germany. [7]NORMENT Centre, Institute of Clinical Medicine, University of Oslo and Division of Mental Health and Addiction, Oslo University Hospital, Oslo, Norway. [8]Department of Psychology, University of Oslo, Oslo, Norway. [9]Department of Psychology, PROMENTA Research Centre, University of Oslo, Oslo, Norway. [10]These authors jointly supervised this work: Havdahl, A., Pingault J-B. ✉e-mail: a.allegrini@ucl.ac.uk

neuropsychiatric risk across generations reflects a complex interplay between genetic and environmental risks. We can specify two broad types of intergenerational effects, related to so-called 'direct' and 'indirect' genetic pathways of transmission within families. First, direct genetic effects originate in an individual's genome. They follow from the genetic transmission of risk, whereby parents transmit their genetic predispositions to their offspring, which in turn relate to specific individual's phenotypes (e.g. genetic variants contributing to anxiety are transmitted to the offspring and, then, directly contribute to offspring anxiety). Second, indirect genetic effects originate in the parent genome, but are independent of the child genotype, and are thought to be accounted for by the family environment (e.g. genetic variants, even when non-transmitted to the offspring, can affect parental depression, which may in turn affect parenting style and, thus, indirectly, child anxiety).

The availability of large family-based genotyped cohorts allows us to untangle direct and indirect genetic effects. This can be done by jointly modelling parent and offspring genetic predispositions, as indexed by polygenic scores (PGS). PGS are predictors of the genetic predisposition of an individual for traits of interest, e.g. the polygenic score for depression. In this context, an investigation of how neuropsychiatric-related predispositions are associated with childhood emotional and behavioural difficulties can be especially useful in at least two ways. First, we can obtain unbiased estimates of 'direct' polygenic risk contributions. Having unbiased estimates of direct effects as a reference can be informative for most studies that typically only comprise singletons and thus cannot take advantage of designs relying on family structure. In such studies of singletons, estimates of associations between PGS and outcomes can be inflated, or completely accounted for, by indirect genetic effects[3]. Indirect genetic effects are generally understood as effects that originate in another individual's genome—typically family members, like parents in the present study— and are mediated via environmental processes independent of genetic transmission. The parental genome thus *indirectly* impacts offspring outcomes. Indirect genetic effects are often referred to as *genetic nurture* due to the hypothesized role of nurturing in this transmission pathway (e.g. parental genetics shapes nurturing practices within the family like parenting, which in turn impacts childhood outcomes). In addition to genetic nurture per se, indirect genetic effects can also reflect other genuine indirect effects that do not directly involve within-family nurturing (e.g. parental income affects the choice of neighbourhood or school which impacts child outcomes). Importantly, however, estimates of indirect effects can also reflect biases from genetic and environmental confounding[3], including assortative mating[4] and social stratification[5]. Irrespective of what they are comprised of, indirect genetic effects will bias between-family estimates of genotype–phenotype associations[6], but can be adjusted for in within-family analyses to recover direct genetic effects, such as in the trio design via estimation of PGS-phenotype effects conditional on the parental PGS.

Second, jointly modelling parent-offspring PGS in trio models allows us to gain insights into specific risk factors responsible for indirect genetic contributions to child emotional and behavioural difficulties. For example, indirect genetic contributions to child depression may be explained by parental risk factors beyond depression, such as parental anxiety and neuroticism, suggesting specific environmental risk factors for child depression[7,8].

Most of the polygenic score work in this area has been spearheaded by studies investigating educational outcomes[9,10], where indirect genetic effects were found to account for about half the size of the PGS effect on education[11]. More recently mixed evidence is starting to accumulate in childhood psychiatry[12–15]. Here, we leverage genomic data from 14,959 family trios to investigate shared vs specific effects of parental neuropsychiatric-related risk on emotional and behavioural difficulties in childhood.

Emotional and behavioural difficulties co-occur in complex ways throughout development[16,17], highlighting the importance of investigating shared and specific predictors. In this context, the latent variable modelling framework allows for the investigation of shared variation across emotional and behavioural difficulties. Within this framework, the co-occurrence of symptoms can be organized in hierarchical structures where shared variance across observed variables (such as observed difficulties) is captured by latent dimensions reflecting phenotypic domains. Domains are more specific at lower levels (e.g. a 'depression' domain), and more general at higher levels (e.g. an 'internalizing' domain, reflecting the common variance across, e.g., depression and anxiety). Typically, a common feature of hierarchical models is that variance shared across all domains, or the entirety of observed traits, is captured by a unique general domain, commonly referred to as 'the general psychopathology factor' (or '*p*-factor'[18]). Within a common cause framework, this general domain is purported to explain why neuropsychiatric conditions and symptoms tend to co-occur.

Every statistical model inherently makes assumptions about the data and their structure. For example, different hierarchical models imply distinct mechanisms through which higher-order domains are linked to observed traits, such as emotional and behavioural symptoms[19]. If taken at face value, a corollary of such models is that risk factors, such as parental neuropsychiatric conditions, are assumed to link to specific symptoms through broader domains. For example, parental neuroticism could affect children's general predisposition to emotional and behavioural difficulties (*p*-factor) which, in turn, impacts narrower domains such as depression, as well as specific depression symptoms such as fatigue and low mood [Box 1]. However, risk factors may be unique to specific symptoms, or express heterogeneously at different levels of the hierarchy suggesting a more nuanced picture (e.g. ref. 20). For example, risk factors for depression do not consistently associate across depressive symptoms indicating heterogeneity in this (latent) construct[21]. In this context, it is important to investigate how risk factors relate to the building blocks of the hierarchical structure of neuropsychiatric conditions. We can bring this idea in the context of childhood emotional and behavioural difficulties by examining how PGS for neuropsychiatric-related traits affect different levels of the hierarchical structure, i.e. affecting specific symptoms (e.g., easily distracted), specific domains (e.g., inattention), or a unique general domain (*p*-factor).

Previous work has investigated how individual predispositions to neuropsychiatric conditions, indexed by PGS for neuropsychiatric (and related) traits, are associated with general and specific emotional and behavioural difficulties in childhood. This body of research generally suggests largely transdiagnostic, non-specific effects[22,23], albeit with more nuance depending on the modelling strategy adopted and the phenotypic domain tapped into by a given PGS[24]. A limitation of this body of research is the reliance on samples of singletons. Here, we implement family-based analyses using genotyped trios, enabling the estimation of both direct and indirect genetic effects on specific versus general domains of child emotional and behavioural difficulties, with implications for interventions. For example, if parental predisposition to depression shows specific links with child depressive symptoms, treatment of parental depression may be likely to impact only child depressive symptoms rather than emotional and behavioural difficulties more generally.

In summary, we leverage data from family trios, using PGS to investigate the intergenerational transmission of neuropsychiatric-related predispositions. First, we examine the direct effects of children's polygenic neuropsychiatric-related predispositions on general and specific emotional and behavioural domains (i.e. adjusted for confounding by indirect genetic effects). Second, we examine parental indirect genetic effects on general and specific emotional and behavioural domains. Finally, we test whether direct and indirect genetic

## BOX 1

# Relationship between risk factors and symptoms of behavioural and emotional difficulties

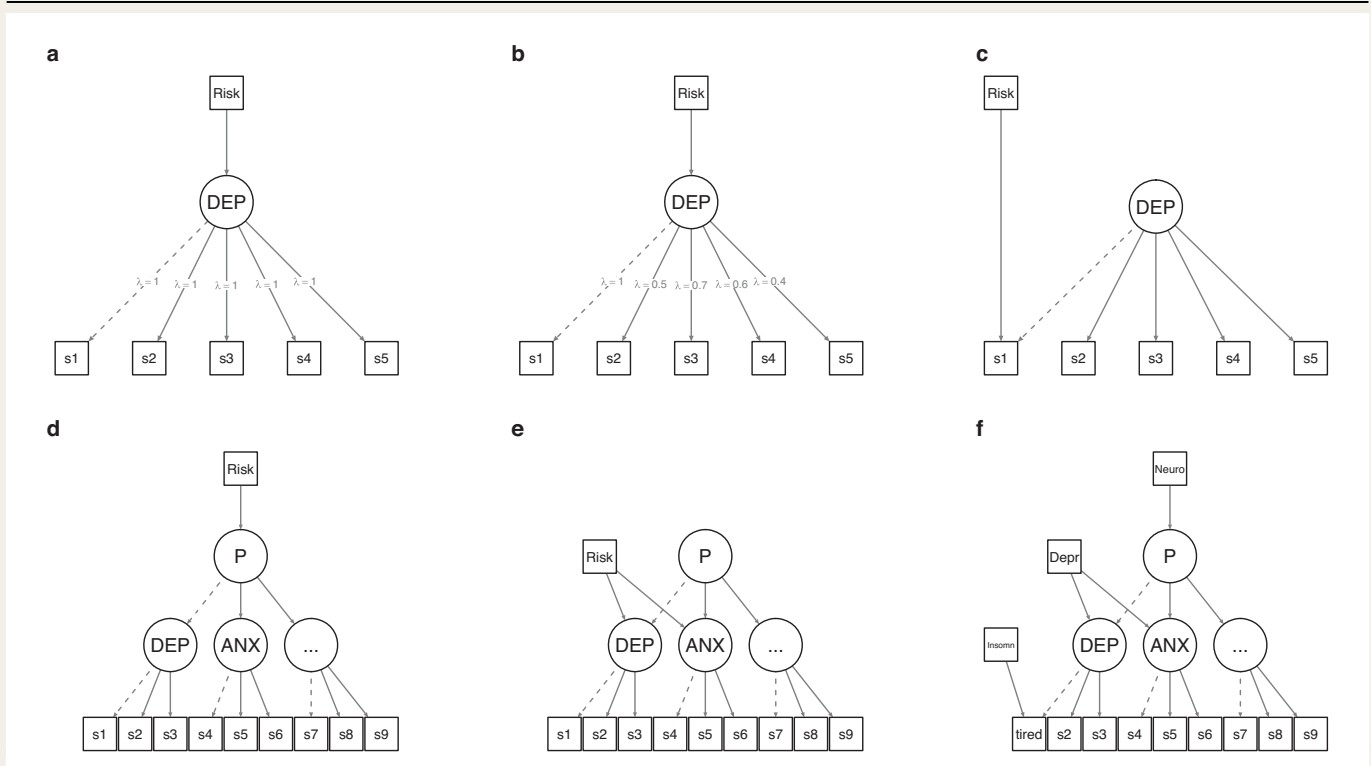

*DEP* depression domain, *ANX* anxiety domain, *P* psychopathology domain, *S1–S9* symptoms 1–9, *Neuro* neuroticism, *Depr* depression, *Insomn* insomnia.

Typically, risk factors for neuropsychiatric phenotypes, such as emotional and behavioural difficulties, are assessed by testing their associations with sum scores aggregating across a number of symptoms. This modelling framework is statistically equivalent to a structural equation model (SEM) where a latent construct, say depression, is defined by symptom indicators all contributing the same to this latent construct (panel a)[38]. When fitting this model, we are then implicitly assuming that the effects of a given external risk factor (say maternal depression) on different depressive symptoms are: 1. completely mediated by this latent construct, which in turn is causing the symptoms and 2. the (mediated) effects of the risk factor on the symptoms are equivalent. Depending on the construct of interest, this may be implausible in practice for any given risk factor, as shown elsewhere for depression[21]. For example, a particular risk factor may be mediated by the latent construct depression and have different effects on different symptoms. This can be represented by an SEM model where different symptoms load differently on the latent construct (panel b). However, risk factors may also be unique to particular symptoms. For example, a predisposition to insomnia may affect fatigue, a symptom of depression, independently of depression (panel c). This relationship may however be obscured if we only considered the latent construct

depression (or a sum score thereof) as our level of analysis. In this context considering different levels of analysis is important to understand shared vs unique contributions of a particular risk factor to symptom clusters.

In a similar fashion, we can investigate how particular risk factors relate to different domains of emotional and behavioural difficulties. Panels 'd–f' depict the relationship between different symptom domains as a hierarchical model where a general domain captures shared variance across lower-level specific domains. Similar to the case of depression, we can conceptualize external risk factors as having shared effects across all symptom domains of a given hierarchy (panel d) or only across a subset of such domains (panel e). For example, parental risk for neuroticism may affect different emotional and behavioural domains, consistent with mediation by a general psychopathology domain ('P'). Conversely, parental risk for depression may only be relevant for a subset of domains, for example, depression and anxiety, but not for others. In turn, this depression risk factor may exert effects that generalize across all depressive symptoms. Concurrently, risk factors for particular symptoms might be at play (say an individual predisposition to insomnia may affect fatigue as in the example above). These effects are non-mutually exclusive, but are likely to be obscured or confounded when using only sum scores or just one level of analysis.

effects on child emotional and behavioural difficulties are general, domain-specific, or symptom-specific in nature.

## Results

The sample comprised 14,959 genotyped unrelated family trios from Norway, for which at least one phenotypic observation measured when

children were aged 8 years was available ("Methods"). As a baseline for our analyses, we fit a second-order hierarchical model to item-level data on emotional and behavioural difficulties, specifying domains corresponding to each of the six scales from which items originated (top panel Fig. 1). These six symptom scales comprised maternal-rated measures of children's depression and anxiety symptoms, conduct

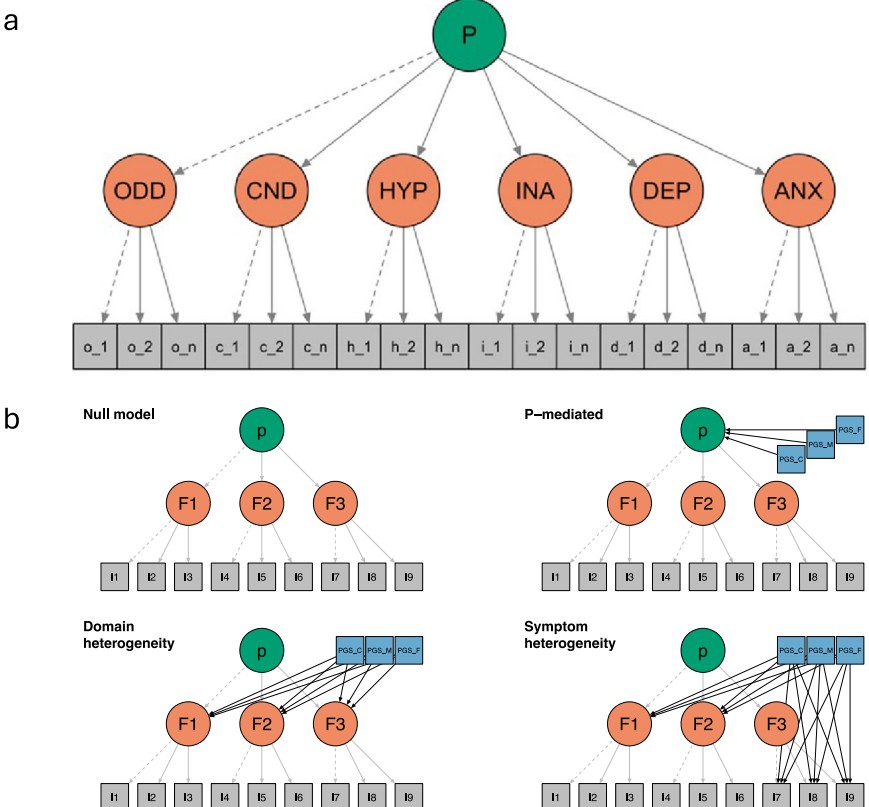

**Fig. 1 | Vignettes of models tested. a** Second-order model fit to symptom-level data across six emotional and behavioural difficulties scales. **b** Four models tested for each polygenic score: null model (polygenic scores are not associated with any outcome), P-mediated model, domain-heterogeneity model and symptom-heterogeneity model. ODD oppositional-defiant domain, CND conduct domain, HYP hyperactivity domain, INA inattention domain, DEP depression domain, ANX anxiety domain.

problems, oppositional-defiant behaviours, hyperactivity and attention levels. Supplementary Data 1 reports frequency distributions for the item-level data. In the baseline model, each symptom was allowed to load onto one of six first-order domains corresponding to the six scales of emotional and behavioural difficulties under study. Variance in common across all first-order domains was further summarised by a unique second-order general emotional and behavioural difficulties domain (P). The baseline model fitted the data well (Supplementary Data 2 and 3).

To assess power for our analyses under different scenarios, we used weights from this model to simulate data and run power simulations based on combinations of three data-generating mechanisms (Supplementary Material). These involved joint effects of PGS for parents and offspring over the domain-general (P) or the domain-specific factors. Simulations showed that we had at least 80% power to detect a small indirect genetic effect of beta = 0.03 when the data-generating mechanism involved effects completely mediated by either the general or specific domains (Figs. S1–S3).

For our main analyses, we calculated polygenic scores for parents and offspring to test their conditional effects over the general and specific domains (conditional models, adjusted for all three PGS— mother, father, offspring), and compared them to estimates obtained from models where PGS effects were considered separately for parents and offspring (unconditional models, i.e., not adjusted for the PGS of other family members). To this end, we generated PGS from genome-wide association study (GWAS) summary statistics for eight psychiatric and neurodevelopmental conditions: Attention-deficit hyperactivity disorder (ADHD), Autism spectrum disorder (AUT), Bipolar disorder (BIP), Schizophrenia (SCZ), Anorexia nervosa (AN), Anxiety (ANX),

Post-traumatic stress disorder (PTSD), and Broad depression (DEP). Emotional liability, sleep problems and pain are common across many of these conditions, and PGS for these traits have been previously found to be associated with general and specific emotional and behavioural domains in independent samples[22,24]. We thus also generated PGS for Neuroticism, Insomnia and Chronic pain (CPAIN). In addition, we calculated two multivariate PGS obtained from the first unrotated principal component of the neuropsychiatric PGS ('poly-genic-P') and from all the scores together ('PC1' PGS) (Supplementary Data 4 for the principal component analyses weights). These two different PCA PGS were generated as a sensitivity analysis to first test whether polygenic-p contributed mainly to the general domain (P), and second, whether a multivariate PGS extended beyond neuropsychiatric traits alone made similar contributions. Finally, we generated a PGS based on a GWAS of hair colour (red) as a negative control in our analyses.

**Direct genetic effects**

Among the 14 conditional models, each corresponding to one PGS (e.g. DEP), 6 were favoured over a null model, suggesting that, together, the PGS for the father, mother and offspring for the corresponding trait were predictive of the outcomes, as captured by the general (P) domain in the second-order model. Figure 2 displays results for those 6 models, i.e. beta estimates (unstandardized solution) and corresponding CIs from conditional models (independent effects of parent-offspring PGS). These included ADHD, autism, depression, neuroticism and chronic pain PGSs, as well as both multivariate PGS (PC1 not shown, Fig. S4 shows results across all PGS). Our negative control was not favoured over the null model. Supplementary Data 5 and 6 report results for all conditional and unconditional models.

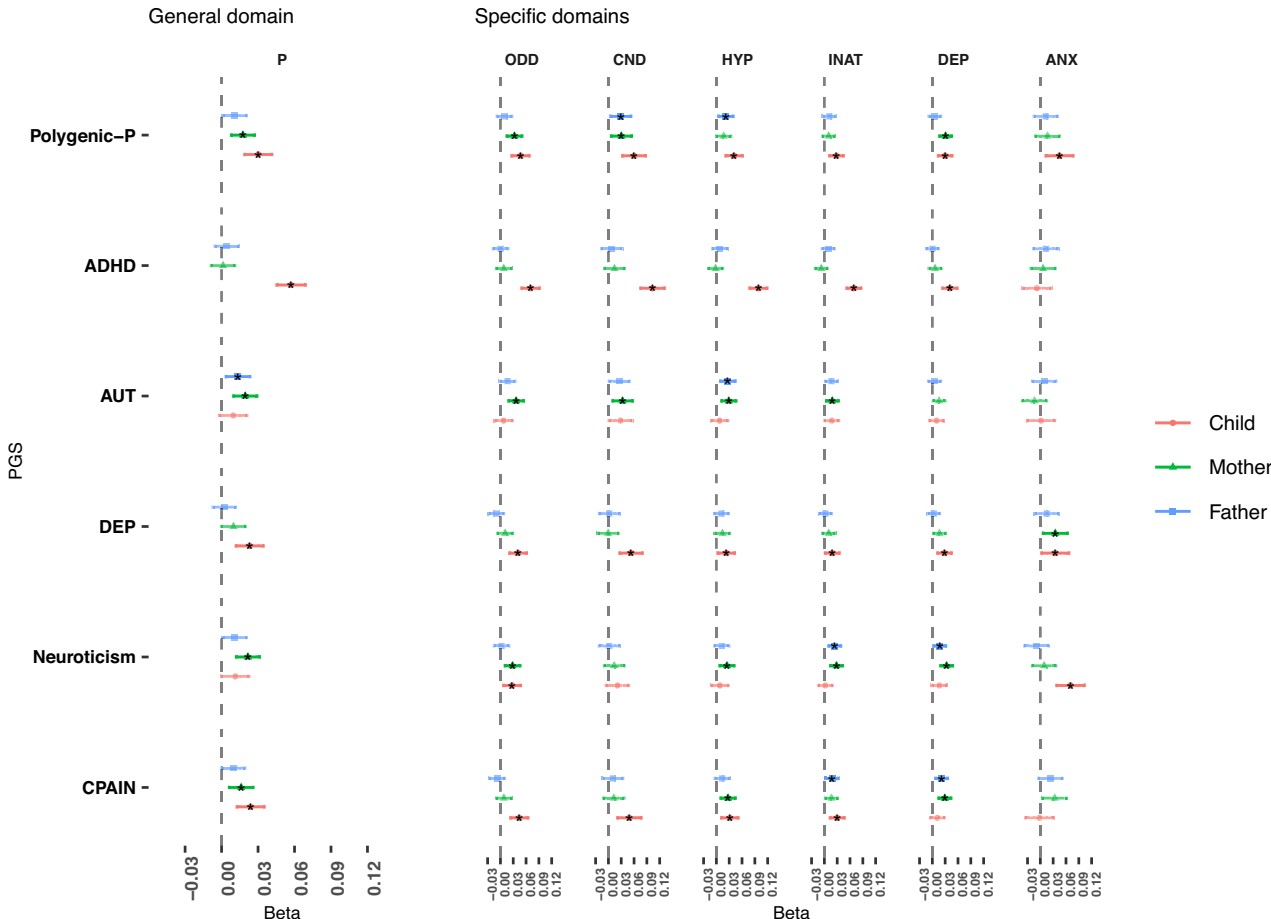

**Fig. 2 | Parent-offspring PGS effects on general and specific emotional and behavioural difficulties domains.** Effect sizes and confidence intervals for conditional models considering parent and offspring polygenic scores (PGS) effects on the general (P) and specific emotional and behavioural difficulty domains. Sample size N = 14,959. Point estimates represent beta coefficients and error bars are 95%

CIs. ADHD Attention-deficit/hyperactivity disorder, AUT Autism spectrum disorder, DEP Broad depression, CPAIN Chronic pain, polygenic-P first unrotated principal component of all neuropsychiatric PGS. Facets: ODD Oppositional defiant disorder, CND Conduct disorder, HYP Hyperactivity, INA Inattention, DEP Depression, ANX Anxiety. *Survives correction for multiple testing ("Methods").

Direct genetic effects (i.e. child effects in conditional models represented in Fig. 2) were observed across all emotional and behavioural difficulties domains using PGS for polygenic-P (standardized beta range = 0.043–0.075), depression (std beta = 0.028–0.062), and ADHD (std beta = 0.071–0.131, with the exception of the anxiety domain). Direct genetic effects with more specificity were also identified. For example, effects of PGS for chronic pain (std beta = 0.041–0.058) were evident in all domains except on emotional problems, and the neuroticism PGS associated only with anxiety (std beta = 0.078) and oppositional-defiant domains (std beta = 0.034).

## Indirect genetic effects

Several indirect (paternal and/or maternal) genetic effects were also evident. Supplementary Data 7 reports standardized and unstandardized coefficients and inferential statistics for all PGS models. For example, maternal indirect effects of polygenic-P were evident in the general domain (beta = 0.017, se = 0.004, p = 3.33e-4, std beta = 0.041), as well as parental (either maternal, paternal or both) indirect genetic effects across most specific domains, with the exception of the inattention and anxiety domains. In contrast, for example, no parental effects of the ADHD PGS were observed on any outcome. Maternal indirect genetic effects on P were also observed for the chronic pain PGS (beta = 0.016, se = 0.005, p = 1.39e-3, std beta = 0.037). Similarly, parental indirect effects on the general domain P were observed for the neuroticism and autism PGS, notably in the absence of evidence for

direct genetic effects. Indirect genetic contributions to a number of specific domains were observed across all these PGS (Fig. 2 and Supplementary Data 7).

## Polygenic score contributions across domains

Figure 3 (top panel) shows a comparison of standardized regression coefficients between the conditional and unconditional models for the offspring PGS. The figure shows that the strongest polygenic predictor of childhood emotional and behavioural difficulties across all domains was ADHD PGS. Notably, an exception to this trend was the anxiety domain, for which the neuroticism PGS was the strongest predictor. Polygenic-P, along with depression and chronic pain PGS also emerged as consistent predictors across the board. By contrast, for parental indirect effects, the strongest predictors tended to be the PGS for autism, polygenic-P, and neuroticism, albeit with a stronger signal overall for the maternal effects compared to the paternal effects (Supplementary Fig. S5). In addition, a general trend to attenuation in conditional models (i.e., most points sitting below the diagonal) can be observed across PGS traits, with the conditional models typically yielding lower (shrunk) standardized estimates than the unconditional model.

## Effect size shrinkage

Effect size estimates of parent and offspring PGS are expected to differ between conditional and unconditional models (Supplementary

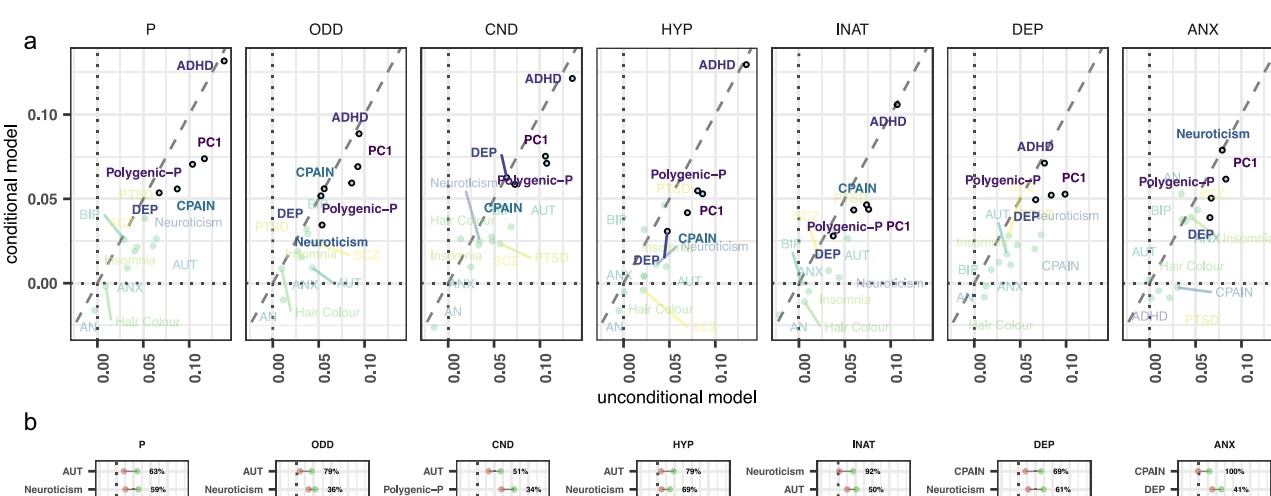

**Fig. 3 | Polygenic scores contributions across emotional and behavioural difficulties domains. a** Comparison of standardized regression coefficients for offspring PGS effects from conditional to unconditional models, showing the relative importance of PGS contributions across emotional and behavioural difficulties. Faded: does not survive correction for multiple testing or not selected over the null model. **b** Shrinkage of standardized effects for the offspring PGS from unconditional (i.e. green) to conditional (i.e. red) models across emotional and behavioural difficulties domains, restricted to models favoured over the null model (PC1 PGS not shown for clarity). Estimates are plotted in descending order of shrinkage.

Note: Shrinkage estimates in (**b**) were obtained from standardized estimates for the child PGS in the conditional and unconditional models shown in panel a (see "Methods"). ADHD Attention-deficit/hyperactivity disorder, AUT Autism spectrum disorder, BIP Bipolar disorder, SCZ Schizophrenia, AN Anorexia nervosa, ANX Anxiety, PTSD Post-traumatic stress disorder, DEP Broad depression, CPAIN Chronic pain, PC1 first unrotated principal component of all neuropsychiatric (and related) PGS, Polygenic-P first unrotated principal component of all neuropsychiatric PGS. Facets: ODD Oppositional defiant disorder, CND Conduct disorder, HYP Hyperactivity, INA Inattention, DEP Depression, ANX Anxiety.

---

**BOX 2**

# Summary of the analytical procedure

**for each** PGS:
1. fit *null model*
2. fit alternative models (*p-mediated* and *domain-heterogeneity* models)
3. compare models: $\chi^2_{diff}$ (*null, p-mediated, domain-heterogeneity*)
4. **if** (fit *p-mediated* > fit *null* model) **then**
5. statistical inference (*p-mediated* and *domain-heterogeneity* models)
6. **if** (fit *domain-heterogeneity* > fit *p-mediated* > fit *null*) **then**
7. **for each** specific factor in the *domain-heterogeneity* model:
8. fit *symptom-heterogeneity* model
9. compare models: $\chi^2_{diff}$ (*domain-heterogeneity, symptom-heterogeneity*).
10. **if** (fit *symptom-heterogeneity* > fit *domain-heterogeneity*) **then**
11. statistical inference (*symptom-heterogeneity model*)

Data 7–9) because direct and indirect paths of genetic transmission can be reciprocally confounded. For example, demographic phenomena, captured by indirect genetic effects (proxied by parental PGS effects), can contribute to PGS–phenotype associations in the offspring. Vice versa, when not adjusted, parental PGS effects on offspring phenotypes will also capture genetic transmission, not only putative genetic nurture effects. We can quantify the extent to which

this is the case by considering the relative shrinkage of parameter estimates from unconditional to conditional PGS models ("Methods"). Figure 3 (bottom panel) depicts shrinkage of offspring PGS effects, for those models favoured over the null model ("Methods" section and Supplementary Data 10; PC1 PGS not shown for clarity).

For example, virtually no shrinkage was observed for the ADHD PGS across domains, indicating that only direct genetic effects were contributing to this relationship. Conversely, the effects of the Autism PGS underwent substantial shrinkage suggesting either indirect genetic effects or demographic confounding at play. With the exception of the ADHD PGS, virtually all offspring PGS effects underwent shrinkage. Taking for example the polygenic-P PGS, shrinkage was evident across all specific domains (range = 25–38%).

**Polygenic risk transmission to symptoms of emotional and behavioural difficulties**

We then proceeded to test formally whether polygenic contributions of neuropsychiatric-related risk across symptoms of emotional and behavioural difficulties were likely to be mediated by the general or specific domains, or whether effects were heterogeneous across symptoms within domains. Our analytical strategy is detailed in the "Methods" and summarized in Box 2. Briefly, for each PGS model favoured over the null model, we statistically compared a restrictive common pathway model to a more flexible specific pathways model[25,26], at both levels of the hierarchy (i.e., first- and second-order level domains). The more restrictive model (henceforth 'P-mediated' model) assumes that PGS effects were uniquely mediated by the general domain 'P'. This was compared to a less restrictive model in which effects were freely estimated over specific dimensions (the 'domain-heterogeneity' model). Finally, this latter model was compared to a

model allowing for PGS effects directly over symptom indicators ('symptom-heterogeneity' model), for each specific domain in turn.

Figure 4 shows results for the heterogeneity analyses. For instance, the P-mediated model was favoured for the polygenic-P PGS, indicating that PGS contributions for general neuropsychiatric risk over emotional and behavioural difficulties domains were consistent with mediation by the general domain. The P-mediated model was favoured also for depression and autism PGS. One important distinction, however, is that while for the polygenic-P PGS parental and offspring effects were both observed, parental depression PGS effects were overall near 0 and not statistically significant. This suggests that the mediated effects of the depression PGS over the specific domains were driven mainly by direct genetic effects. Conversely, for the autism PGS, only parental PGS contributions were evident across domains.

As a comparison, for the ADHD PGS a more nuanced picture emerged: The domain-heterogeneity model was favoured for the conduct, inattention and depression domains, suggesting that ADHD PGS effects over symptoms for these traits were likely to be mediated by their corresponding specific domains (i.e. scale-level factors). In other words, there was no symptom heterogeneity at play in terms of (direct) PGS effects. However, for oppositional-defiant and hyperactivity domains, the symptom-heterogeneity model was favoured, suggesting heterogeneity in direct effects over symptoms. In Supplementary Material, we discuss specific examples of symptom and domain-level heterogeneity (Supplementary Data 12 and 13 and Figs. S6–S9).

## Discussion

We investigated direct and indirect polygenic risk contributions of neuropsychiatric-related traits to general and specific domains of childhood emotional and behavioural difficulties. Three main findings emerged. First, we observed that, when adjusting for parental polygenic scores, the offspring polygenic score effects were generally smaller than what would be naively obtained from unadjusted analyses in singletons. Second, parental indirect genetic effects, reflecting polygenic risk for neuropsychiatric traits, were evident across general and specific domains of emotional and behavioural difficulties. Third, overall, indirect genetic contributions tended to be mediated either by the general domain or by the specific domains, while direct genetic effects were also found to contribute heterogeneously across symptoms within specific domains.

An important result for PGS work in developmental psychology regards the decrease, or shrinkage, in effect sizes of PGS–phenotypes associations that is observed when conditioning on parental PGS, hence adjusting for indirect genetic effects. While this phenomenon is well documented for cognitive-related traits[11,27] our results indicate it goes beyond the cognitive domain. For example, our results for the depression PGS provide converging evidence with previous GWAS work[28]. However, we also show that shrinkage depends on the target phenotype of interest, suggesting a more nuanced picture than generalized confounding. For example, the neuroticism PGS effects on the depression domain exhibited substantial shrinkage, but this was not the case with the anxiety domain, where no shrinkage was observed.

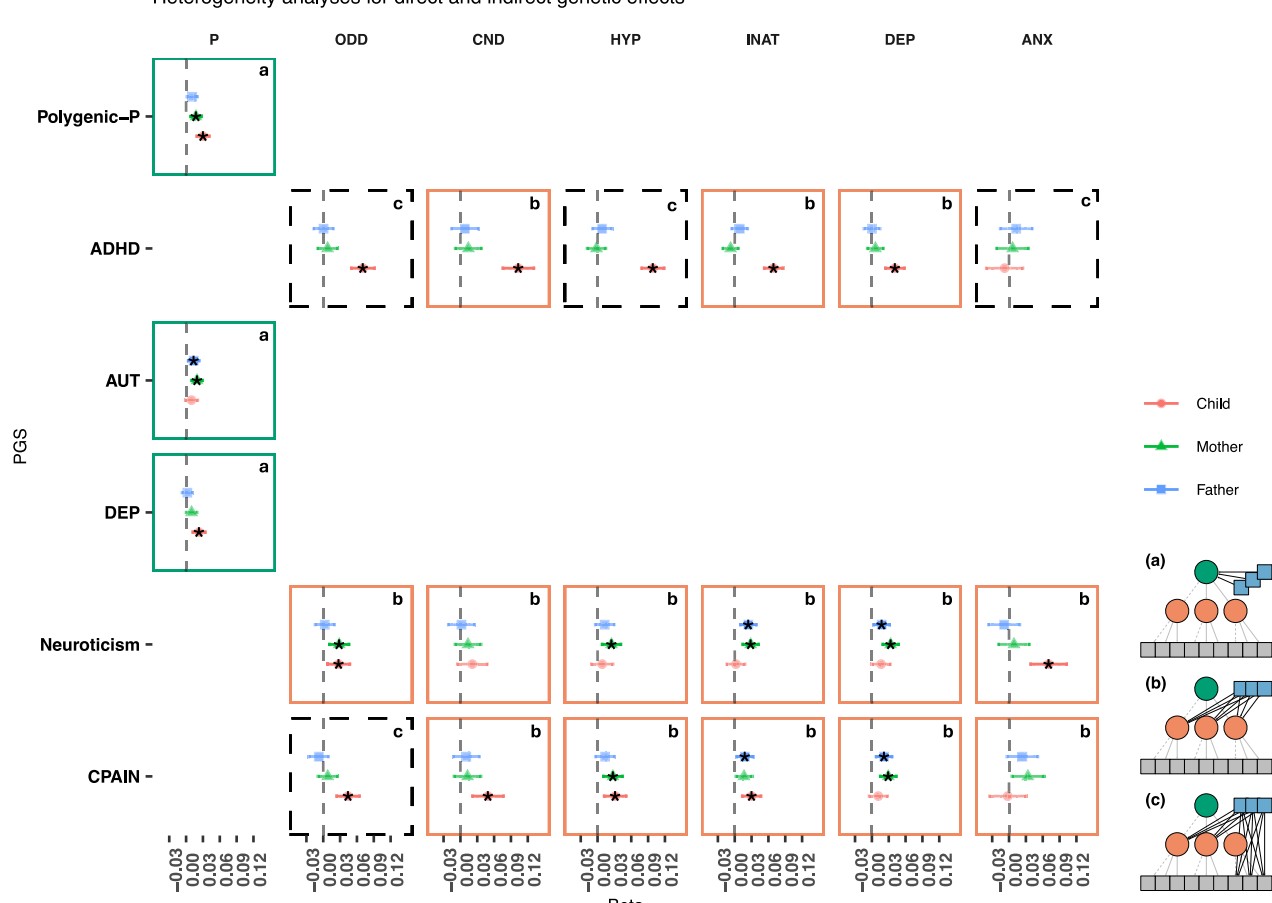

**Fig. 4 | Heterogeneity test results across polygenic score models.** Facets contour and letters indicate the favoured model: a/Green = P-mediated model, b/orange = domain-heterogeneity, c/dashed-black = symptom heterogeneity model. ADHD Attention-deficit/hyperactivity disorder, AUT Autism spectrum disorder, DEP Broad depression, CPAIN Chronic pain, Polygenic-p first unrotated principal component of all neuropsychiatric PGS. Facets: ODD Oppositional defiant disorder, CND Conduct disorder, HYP Hyperactivity, INA Inattention, DEP Depression, ANX Anxiety.

This suggests that while environmentally mediated effects related to neuroticism may play a role in childhood depression, the main reason for the association with childhood anxiety is genetic overlap.

Our findings triangulate evidence on genetic nurture effects from previous work using different methodologies in the MoBa sample. For example, [7]showed genetic nurture effects on depressive symptoms (but not anxiety), while[29] found only suggestive evidence in this regard. Here, we find evidence that parental risk is linked, albeit not strongly, with both anxiety and depression symptomatology in childhood. Indirect genetic effects on anxiety were driven by parental genetic risk for depression, while indirect genetic effects on depression were driven by neuroticism, chronic pain, general polygenic neuropsychiatric risk (polygenic-P) and, to a lesser extent, by depression itself.

Recent work using non-transmitted polygenic scores for neuropsychiatric conditions has found limited evidence for indirect genetic effects on emotional problems across development[30]. The discrepancy in findings between our study and Shakeshaft et al.'s study may be due to differences in power, arising from different sample sizes and designs. We note that the sample size for Shakeshaft et al. was about half the sample size of the present study, potentially limiting its ability to detect genetic nurture effects (consistent with power calculations presented in the Supplementary Material, page 4). In addition, the trio design that we employed is more powerful than the non-transmitted approach to detect indirect genetic effects[31].

Other converging evidence emerging from our analyses involved indirect genetic effects of the neuroticism PGS on the opposition-defiant symptom domain. Evidence employing different methodologies, such as variance decomposition-based methods, is starting to accumulate implying genetic nurture effects over the externalizing spectrum[32,33]. Interestingly, however, while both maternal and paternal indirect genetic effects were evident when considering neuroticism and chronic pain contributions to the depression domain, only maternal genetic effects were evident for neuroticism and ODD. This might suggest a form of rater bias effect whereby mothers with a higher genetic predisposition to neuroticism are also more likely to report more child oppositional and defiant behaviour.

We previously did not find evidence for genetic nurture effects on the externalizing spectrum using similar methodology in the MoBa sample[12,13], except for neuroticism on ADHD. Conversely, here, we observed parental indirect genetic effects of autism and general neuropsychiatric risk on the conduct domain. Furthermore, we find autism, neuroticism, chronic pain and general neuropsychiatric risk PGS effects on the hyperactivity and inattention domains. On the one hand, the PGS for autism, chronic pain and general neuropsychiatric risk were not previously investigated in this context. On the other, it might be that separating between domain subdimensions, such as inattention and hyperactivity, within a multivariate setting is a useful level of analysis in this regard. It is of note that for several models involving polygenic predispositions for chronic pain, neuroticism and autism, only parental contributions were evident—suggesting either family-wide environmental processes as a main route for transmission, or more unspecific confounding driving this relationship. Future studies should disentangle these alternative hypotheses.

Finally, we provide an indication of how polygenic scores may differentially link to behavioural and emotional symptoms. We found that the ADHD PGS was not a specific predictor restricted to related domains (hyperactivity and inattention), but was the strongest predictor across emotional and behavioural difficulties including general and specific domains. While comparisons of PGS effect sizes is hampered by considerations of GWAS power, we note that the ADHD GWAS was not the most powered across those considered. The association patterns of other PGS, including chronic pain, depression and polygenic-P were also largely unspecific (i.e. not unique to any one domain), except for the anxiety dimension. These results are consistent with previous recounts in independent child cohorts[22–24].

We cannot exclude that these results may partly arise due to cross-trait assortative mating[34], and/or the phenotypic distance between child phenotypes and some of the adult GWAS employed here. These considerations may impact the level of specificity of PGS and explain the observed attenuation of PGS effects. However, we note that shrinkage was not observed selectively for adult-based GWAS (e.g. depression and neuroticism) compared to child-based GWAS (e.g. ADHD and ASD). After conditioning on the parental PGS, we observed that the PGS for ADHD, neuroticism and chronic pain was associated heterogeneously across specific domains and symptoms, suggesting some degree of specificity.

Such specificity of direct and indirect effects can have practical implications for research and intervention strategies. Considering direct genetic effects, we show that focusing on the symptom level in addition to the sum score level can provide novel insights. For example, our findings indicate that polygenic predisposition for ADHD relates more to motor rather than verbal impulsivity items (Supplementary Material). By extension, this may point to different aetiologies for subdimensions of hyperactivity-impulsivity in childhood ADHD having to do with 'motion' versus 'speech'. Distinguishing these subdimensions could yield further nuance in child developmental psychology work as well as genomic studies[35,36]. Considering indirect genetic effects, specificity might point to interventions aimed at parental characteristics that are likely to have an impact on particular symptom domains in their offspring. As an example, paternal and maternal indirect effects of neuroticism were more important for depression and inattention (Fig. S9), suggesting that interventions aimed at reducing neuroticism in the parents may have an impact on childhood symptoms of depression and inattention, but not necessarily on other symptom domains. Conversely, the maternal and paternal indirect effects of the polygenic-P PGS were found to be mediated by the general domain, which can be considered as an index of comorbidity (as shown empirically elsewhere, Fried et al.[37]). This suggests that indirect genetic effects (i.e., putative environmental, family-wide processes) related to general parental psychopathology may impact the comorbidity of symptoms in childhood, and may thus be a target of preventive interventions and improve prognosis across a range of emotional and behavioural difficulties.

The results of the present work need to be considered in light of a number of caveats. First, GWAS of behavioural traits, based on scale-level measurements, can also be considered as (implicit) common pathway models (with equal indicator weights). That is, GWAS of sum scores can be likened to testing the hypothesis that an SNP is influencing all indicators via the same route (namely the scale-level factor) and with all indicators equally contributing to the factor[38,39]. Symptom-level PGS obtained from item-level GWAS (e.g. ref. [40]) are therefore bound to better capture specificity, and might be promising avenues to fine-tune the search for heterogeneous pathways of risk, whether for direct or indirect genetic effects. Second, the fact that several PGS were not favoured over the null model, in conditional analyses, does not mean that they are not worth investigating further. We should expect that at larger sample sizes even more modest effects will be uncovered. Third, replication of these findings across different raters is essential. As mentioned, some of the indirect genetic effects we detected, for example, parental PGS effects of chronic pain on the hyperactivity domain, might have arisen because of rater bias. For example, mothers with a higher genetic predisposition to chronic pain might be more likely to perceive their offspring as hyperactive and in turn rate their offspring's hyperactivity higher. Relatedly, the child-parent agreement on internalizing symptoms for the SCARED questionnaire is low[41] and this may have impacted our results for the anxiety domain.

Future studies should also examine a broader range of neuropsychiatric outcome measures. We found that autism PGS showed only indirect, and not direct, genetic effects on emotional and

behavioural difficulties. However, there may be more direct genetic effects on core autistic traits such as atypical social communication and restricted and repetitive behaviours. Further, replication of all findings in independent cohorts and validation with different designs (e.g. ref. 5), including causal inference methods such as within-family Mendelian randomisation, is needed.

Furthermore, a number of caveats relating to genetic nurture and specifically to the models we employed here should be addressed in future research. First, the phenotypic definition in GWAS is of key importance. PGS will capture (direct and indirect) average genetic effects on a phenotype reflecting a specific developmental period and milieu. In the context of genetic nurture, the implications of mapping adult-derived PGS on childhood symptomatology are not straightforward. For example, adult-based GWAS of psychiatric traits is unlikely to capture the full complexities of genetic influences on childhood phenotypes across development[42] (e.g. early vs late-onset ADHD). However, GWAS of child phenotypes is unlikely to serve as good proxies of parental risk factors relevant to those child phenotypes (e.g. parental neuroticism). As such, GWAS of relevant adult phenotypes may be more likely to appropriately capture genetic nurture effects (e.g. adult depression GWAS -> parental depression -> parenting -> offspring outcomes). Future GWAS work leveraging within-family designs across the developmental spectrum could improve our understanding in this regard.

Second, rare variation, and in particular the interplay between rare and common variation[43], may play an important role in the risk transmission of psychiatric traits. As more developmental cohorts with sequencing data are becoming available this will be an important avenue to be explored.

Third, assortative mating may account for a substantial proportion of indirect effects detected here and is something that we cannot exclude based on a two-generation model alone[4]. A three-generation model, or other methods based on different assumptions and designs that can further disentangle parental assortment from indirect genetic effects due to genetic nurture will be a key avenue for future work in this area[4,5,44,45]. Finally, future work investigating the degree to which these results replicate across the genetic ancestry spectrum and in diverse social settings is warranted.

In conclusion, by examining family-level genomic data we provided an account of how neuropsychiatric-related polygenic risk contributes to childhood emotional and behavioural difficulties via direct and indirect genetic effects. Overall, we observed evidence consistent with a putative environmental route to domain-general symptomatology, while also demonstrating domain-specific direct and indirect neuropsychiatric-related genetic contributions. An important aim for future studies will be to triangulate this evidence with different designs and independent samples. The goal will be to uncover whether indirect genetic effects detected are in fact attributable to genetic nurture processes, and if so, which specific "nurturing" environments are implicated.

## Methods

### Ethics
Informed consent was obtained from all study participants. The establishment of MoBa and initial data collection was based on a license from the Norwegian Data Protection Agency and approval from The Regional Committees for Medical and Health Research Ethics. The MoBa cohort is currently regulated by the Norwegian Health Registry Act. The current study was approved by The Regional Committees for Medical and Health Research Ethics (ethical approval: 2016/1702).

### Sample
We analyse data from the 'The Norwegian Mother, Father and Child Cohort Study (MoBa)' a population-based pregnancy cohort study conducted by the Norwegian Institute of Public Health[46,47]. Participants

were recruited from all over Norway from 1999 to 2008. The women consented to participation in 41% of the pregnancies. The cohort includes approximately 114.500 children, 95.200 mothers and 75.200 fathers. The current study is based on version 12 of the quality-assured data files released for research in January 2019.

A description of the cohort and the QC performed on genetic data is available elsewhere[48]. Here we focus on a subset of the total cohort comprising complete trios from unrelated families, and further restricted to one child per family for a total of 30,048 families. Selection of individuals among pairs of siblings within families was performed by prioritizing based on phenotypic availability across the six emotional and behavioural domains considered in analyses and described below. That is, within genotyped families with more than one offspring, only one at random was retained, unless phenotypic data was available for only one person in which case that child was prioritized over the others. This yielded a total of 14,959 genotyped family trios with at least one phenotypic observation available across emotional and behavioural difficulties when children were aged 8 years. Of these, 51% of children were females. Figure 5 is a flowchart of the study sample size.

### Measures
**Emotional and behavioural difficulties.** We used maternal reports from questionnaire data collected when children were aged 8 years, the first MoBa wave with an extensive range of measures for emotional and behavioural difficulties. Specifically, we used item-level data measuring symptoms of depression (short mood and feelings questionnaire[49]; 13 items), anxiety (screen for child anxiety-related disorders[50]; 5 items), conduct problems (18 items from the 31 items rating scale for disruptive behaviour disorders [RS-DBD][51]), oppositional defiant disorder (RS-DBD; 8 items), and hyperactivity (RS-DBD; 9 items) and Inattention (RS-DBD; 9 items). Internal consistency for all measures based on items included in the analytical sample is reported in Supplementary Data 11. Supplementary Data 1 reports item frequencies across all measures.

### Polygenic scores
PGS were calculated with LDpred2[52], a Bayesian method to derive polygenic scores using information on the genetic architecture of a trait, and on Linkage Disequilibrium (LD) obtained from a reference panel. To compute PGS, recommended quality control guidelines were followed and variants included were restricted to an extended set of HapMap3 variants[53]. UK Biobank was used as a reference LD panel in PGS calculations using precomputed LD matrices provided in ref. 53. PGS were generated by using the 'auto' option. We generated PGS for a selection of GWAS summary statistics for neuropsychiatric traits including Autism spectrum disorder[54], Bipolar disorder[55], Schizophrenia[56], Attention-deficit/hyperactivity disorder (ADHD)[57], Anorexia nervosa[58], Anxiety[59], Post-traumatic stress disorder (PTSD)[60], Broad depression[61], as well as neuropsychiatric-related traits including Neuroticism[62], Insomnia[63], Chronic pain[64], for a total of 11 PGS. We made this selection in order to cover the phenotypic domains involved in analyses as well as neuropsychiatric-related traits previously found to be associated with general and specific emotional and behavioural domains in independent cohorts[22,24], while being parsimonious in PGS inclusion to limit the burden of multiple testing. In addition, we created two multivariate PGS from the first unrotated principal component of the neuropsychiatric PGS ('polygenic-p') and from all the scores together (PC1) (Supplementary Data 4). Finally, we generated a PGS based on a GWAS of hair colour (red) in UKB as a negative control in our analyses[65]. Supplementary Data 14 reports information on the GWAS summary statistics employed in analyses.

### Analyses
We model item-level data from these measures in hierarchical models of psychopathology. We fit a second-order and a symmetric bifactor

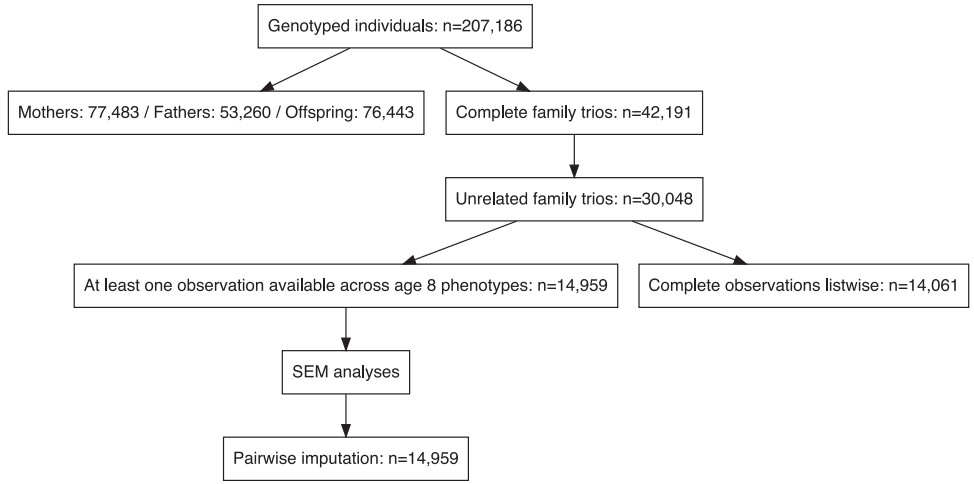

**Fig. 5 | Study flow-chart.** Diagram of the study sample selection.

model (our 'base' models), where items of different emotional and behavioural difficulties scales are loaded on specific latent factors, reflecting their corresponding scales. We then compared model fit using standard model fit indices including CFI, SRMR, RMSEA and chi-square difference test. We examined parameter estimates and model fit indices carrying forward in analyses the more suitable model to the data at hand (Supplementary Material for further discussion)[66].

Importantly, we do not explicitly investigate a particular hierarchical taxonomy either data- or theory-driven (e.g. refs. 67,68), for example, distinguishing between internalizing and externalizing domains. Instead, we focus on how well specific domains, reflecting scale-level measurements of behavioural and emotional difficulties, capture polygenic neuropsychiatric-related risk on corresponding symptomatology. Additionally, we test how well a general behavioural and emotional domain, consistent with a common cause model, captures polygenic effects across specific domains and their symptom indicators.

**Power simulations**

We performed power simulations for a multiple indicators multiple causes (MIMIC) version of the more suitable base hierarchical model. We used weights from the base model to simulate data and run power simulations based on a combination of three data-generating scenarios involving joint effects of parental and offspring polygenic scores on either the general (P) or the specific emotional and behavioural domains (Supplementary Material). Each of these combinations was tested across fixed parameters for the child and mother PGS (beta = 0.04, and beta = 0.03 respectively) and three different parameters for the father polygenic score (betas 0.03, 0.01, and 0.001), holding constant sample size at $N = 15,000$ (our maximum sample size).

**Modelling strategy**

For each PGS considered, we fit a set of structural equation models (SEMs) (depicted in Fig. 1) including a 'null' model where joint PGS effects for parents and offspring traits over the general domain were fixed to 0 ('null model', panel a); and two alternative models where joint parent-offspring PGS effects over either the common factor ('p-mediated', panel b), or the specific factors ('domain-heterogeneity', panel c) were freely estimated. Finally, for each of the specific domains, we estimated a model where PGS effects on items were allowed (i.e. not mediated by general or specific domains; 'symptom-heterogeneity', panel d). Box 2 shows a step-by-step summary of the procedure.

We then performed a model comparison between *null*, *p-mediated*, and *domain-heterogeneity* models, using a chi-square difference test ($\chi^2_{diff}$), as summarised in Box 2. In a similar fashion to

ref. 25 this test provided an indication of whether a common or independent pathway model better fitted the data. In practice, we tested whether direct and indirect PGS effects on behavioural and emotional symptoms were more likely to be mediated by either a general dimension, common across all symptom domains, or a domain-specific dimension (e.g. depression domain). Conversely, if the *symptom-heterogeneity model was favoured* such effects were deemed as heterogeneous across items within domains. All nested comparisons were adjusted for multiple testing, alpha = 0.05/N (N = nested comparisons). If the null model was favoured, no further inference was performed (i.e. fit *p-mediated* < fit *null* model; although we report results for all models tested in Supplementary Data 7–9). If the alternative p-mediated model was not rejected, joint PGS effects over general and specific domains were adjusted for multiple testing (i.e. for all fit *p-mediated* > fit *null* models). P-value adjustment was performed as follows: first, for every offspring polygenic score for which the P-mediated model was favoured over the null model, we tested the hypothesis of direct genetic transmission (i.e. adjusting for indirect effects). To this end, we performed an FDR Benjamini–Hochberg procedure across $N_{tests} = N$ PGS × 7, accounting for all tests performed across the general factor + 6 specific factors. Second, we tested the hypothesis that maternal or paternal indirect effects were present on any given factor, for any given score, conditional on the child polygenic score, performing an FDR Benjamini–Hochberg procedure for a total of $N_{tests} = N$ PGS × 7 × 2 (i.e. further accounting for the fact that there are 2 ways to detect indirect effects, either via the maternal or the paternal PGS). To compare how parent-offspring PGS contributions differed in conditional models compared to unconditional models, we repeated these analyses separately for the offspring PGS (i.e. not adjusting for parental PGS), and for parental PGS (i.e. including both mother and father PGS, but not the offspring PGS in the models). Again, analyses were corrected for multiple testing performing an FDR Benjamini–Hochberg procedure across $N_{tests} = N$ PGS × 7, for the child models, and $N_{tests} = N$ PGS × 7 × 2 for the parent models. Box 2 is a step-by-step summary of the analytical procedure. To quantify the level of shrinkage for parent-offspring PGS effects between unconditional and conditional models we calculated the percentage decrease of standardized beta coefficients between these models, i.e. (1 − (abs(conditional beta)/abs(unconditional beta))) × sign(unconditional Beta) × 100.

All polygenic scores were standardized and adjusted for 20 genetic principal components, genotyping centre, chip and batch, as well as sex and year of birth, and residuals were used in subsequent analyses. Sex and year of birth were also included in all models tested as regressors of the emotional and behavioural items. Biological sex

was derived from chromosomal data. Analyses were not run separately by sex because of power considerations. Analyses were conducted in R (version 3.5.0), RStudio (version 1.4.1717) and on the Colossus HPC Cluster, SEM models were fitted using Lavaan[69] (version 0.6-8). All models were fit using a robust weighted least square (WLS) estimator (WSLMV: diagonally weighted least square estimation with robust standard errors, and mean and variance adjusted test statistics), using pairwise deletion for missing data.

### Reporting summary

Further information on research design is available in the Nature Portfolio Reporting Summary linked to this article.

## Data availability

The consent given by the participants does not open for storage of data on an individual level in repositories or journals. Researchers who want access to data sets for replication should submit an application to datatilgang@fhi.no. Access to data sets requires approval from The Regional Committee for Medical and Health Research Ethics in Norway and an agreement with MoBa.

## Code availability

No custom computer code was used in the study. The software used in the data preparation and analysis were R 3.5, Lavaan 0.6–9, and LDpred2 (bigsnpr[70] 1.12.16). Code for data preparation and analyses is publicly available at https://github.com/AndreAllegrini/IRISK-p[71].

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

## Acknowledgements

This work was funded by the European Research Council (ERC) under the European Union's Horizon 2020 research and innovation programme (grant agreement no. 863981), attributed to J.-B.P. and supporting A.G.A. and L.F. and the UK Research and Innovation (UKRI) under the UK government's Horizon Europe funding guarantee [grant number 575067], attributed to J.-B.P. J.R.B is funded by a Wellcome Trust Sir Henry Wellcome fellowship (grant 215917/Z/19/Z). A.H. is supported by the Research Council of Norway (#274611, #336085, and #288083) and the South-Eastern Norway Regional Health Authority (#2020022, #2021045). OAA is funded by the Research Council of Norway (##324499, #324252, #223273) L.J.H. was supported by the South-Eastern Norway Regional Health Authority (#2022083, #2020023). D.B. was supported by the South-Eastern Norway Regional Health Authority (#2022083). L.H. was supported by the South-Eastern Norway Regional Health Authority (#2020022) and the Research Council of Norway (#336085). W.B. was supported by the Wellcome Trust [224092/Z/21/Z]. The Norwegian Mother, Father and Child Cohort Study is supported by the Norwegian Ministry of Health and Care Services and the Ministry of Education and Research. We are grateful to all the participating families in Norway who take part in this ongoing cohort study. We thank the Norwegian Institute

of Public Health (NIPH) for generating high-quality genomic data. This research is part of the HARVEST collaboration, supported by the Research Council of Norway (#229624). We also thank the NORMENT Centre for providing genotype data, funded by the Research Council of Norway (#223273), South East Norway Versjon 6.9 Health Authorities and Stiftelsen Kristian Gerhard Jebsen. We further thank the Center for Diabetes Research, the University of Bergen for providing genotype data and performing quality control and imputation of the data funded by the ERC AdG project SELECTionPREDISPOSED, Stiftelsen Kristian Gerhard Jebsen, Trond Mohn Foundation, the Research Council of Norway, the Novo Nordisk Foundation, the University of Bergen, and the Western Norway Health Authorities.

## Author contributions

A.G.A. and J.-B.P. conceived and designed the study. A.G.A. conducted the statistical analysis. A.G.A., J.-B.P., and A.H. interpreted the data. A.G.A., L.F., W.B., and L.J.H. contributed to phenotypic and genetic data preparation. A.G.A. and J.B.P. wrote the manuscript with contributions from L.J.H., L.F., W.B., J.R.B., O.A.A., D.B., L.H., and A.H. All authors critically reviewed the manuscript and approved the final version.

## Competing interests

OAA is a consultant to cortechs.ai, and received a speaker's honorarium from Lundbeck, Janssen, and Sunovion. No other authors have competing interests to declare.
