## [Transparent Peer Review file · Nature Communications]

Intergenerational transmission of polygenic predisposition for neuropsychiatric traits on emotional and behavioural difficulties in childhood

Corresponding Author: Dr Andrea Allegrini

Version 1:

Reviewer comments:

Reviewer #3

(Remarks to the Author)

The following is a review of the submission NCOMMS-24-17508A "Intergenerational transmission of polygenic predisposition for neuropsychiatric traits on emotional and behavioural difficulties in childhood". During my review I focused on the authors' adjustments in response to comments by previous reviewers.

Overall, I agree with the positive assessments of the previous reviewers that the manuscript is interesting and methodologically sound. The authors have now substantially improved the manuscript and I believe adequately addressed reviewer comments. My only major comment is that the new Table S14 appears to have introduced some minor mistakes, which should be corrected before publication.

- 1) Table S14 is a very useful addition to the manuscript, but I believe contains a few errors:
 - a) The column SNP h^2 appears to inconsistently show either proportion or percentage. Most values seem to be proportions, but Anxiety and Chronic pain have values above 1.00. Are these percentages? If those could these be expressed consistently as proportion?
 - b) Both hair color and insomnia seem to have the same reference (Watanabe et al., 2019). I believe insomnia should be 2022, instead of 2019.
 - c) Anorexia and PTSD have a range in SNP h^2 . Are these ranges reflecting the different assumed population prevalences? If so, why did the authors decide to use a range of prevalences for these GWASs but single values for the other GWASs?
 - d) Reviewer 1 requested information on "aspects of ascertainment and other demographic and/or clinical characteristics". To provide additional context, I think it would be informative to state whether the GWASs were population-based or case-control, and whether participants were adults or children (Considering also the other reviewer comments 1 regarding using PGS for adult diagnoses in child psychiatry, this could be useful additional information).

Comments not related to previous reviewer comments:

2) SEM estimator: The authors state the use of the DWLS estimator, which is appropriate for ordinal data. Checking the script "02_base_models.r", I see that the default options were used, which also apply robust standard errors and a mean- and variance-adjusted test statistic. I recommend to state this, too, or alternatively refer to WLSMV estimation (DWLS + robust SE), as otherwise readers may think that DWLS was applied with regular standard errors.

3) The authors also refer in modeling strategy to "using pairwise imputation for missing data." While I understand what the authors mean, I believe a more appropriate and common term would be pairwise deletion or pairwise complete analysis.

4) Neumann et al. is once referenced as 2020 and once as 2022, I believe it should be 2022 in both cases.

5) Hair color is mentioned in the manuscript as negative control, but I could not find mentions of the results. Perhaps it would

be good to add a sentence in the results, that the hair color PGS was not associated with child psychopathology (according to my understanding of the supplementary tables).

REVIEWERS' COMMENTS

Reviewer #3 (Remarks to the Author):

The following is a review of the submission NCOMMS-24-17508A "Intergenerational transmission of polygenic predisposition for neuropsychiatric traits on emotional and behavioural difficulties in childhood". During my review I focused on the authors' adjustments in response to comments by previous reviewers.

Overall, I agree with the positive assessments of the previous reviewers that the manuscript is interesting and methodologically sound. The authors have now substantially improved the manuscript and I believe adequately addressed reviewer comments. My only major comment is that the new Table S14 appears to have introduced some minor mistakes, which should be corrected before publication.

We thank the reviewer for the positive assessment and for suggesting helpful edits, below we address their comments point by point.

1) Table S14 is a very useful addition to the manuscript, but I believe contains a few errors:

a) The column SNP h^2 appears to inconsistently show either proportion or percentage. Most values seem to be proportions, but Anxiety and Chronic pain have values above 1.00. Are these percentages? If those could these be expressed consistently as proportion?

We have fixed those typos and now correctly report them as proportions.

b) Both hair color and insomnia seem to have the same reference (Watanabe et al., 2019). I believe insomnia should be 2022, instead of 2019.

Thanks for spotting this, we have now amended the reference with the correct year.

c) Anorexia and PTSD have a range in SNP h^2 . Are these ranges reflecting the different assumed population prevalences? If so, why did the authors decide to use a range of prevalences for these GWASs but single values for the other GWASs?

For each trait we report SNP h^2 estimates and population prevalences as presented in the corresponding publications. In the case of Anorexia and PTSD a range of prevalences and corresponding estimates were provided in the publications. We now clarify this in the Table caption which now reads:

Note. SNP h^2 estimates provided along with population prevalences reflect those reported in the corresponding publications. For case/control traits SNP h^2 is provided on the liability scale. The Insomnia, Depression and Neuroticism sumstats used for PGS calculation did not include the 23&Me sample. SNP h^2 estimates (LDSC) provided for these traits are based on the reduced sample.

d) Reviewer 1 requested information on “aspects of ascertainment and other demographic and/or clinical characteristics”. To provide additional context, I think it would be informative to state whether the GWASs were population-based or case-control, and whether participants were adults or children (Considering also the other reviewer comments 1 regarding using PGS for adult diagnoses in child psychiatry, this could be useful additional information).

Thank you for the suggestion. We have now expanded the table to include more information. Most GWAS were based on a mix of population based and case-control samples. We have included an ‘ascertainment/evaluation of cases’ field, and whether the sample was child vs adult based.

Comments not related to previous reviewer comments:

2) SEM estimator: The authors state the use of the DWLS estimator, which is appropriate for ordinal data. Checking the script “02_base_models.r”, I see that the default options were used, which also apply robust standard errors and a mean- and variance-adjusted test statistic. I recommend to state this, too, or alternatively refer to WLSMV estimation (DWLS + robust SE), as otherwise readers may think that DWLS was applied with regular standard errors.

Thank you for spotting this, we now clarify this point in the methods as follows (page 32 line 24):

All models were fit using a robust weighted least square (WLS) estimator (WLSMV: Diagonally Weighted Least Square estimation with robust standard errors, and a mean and variance adjusted test statistics), using pairwise deletion for missing data.

3) The authors also refer in modeling strategy to “using pairwise imputation for missing data.” While I understand what the authors mean, I believe a more appropriate and common term would be pairwise deletion or pairwise complete analysis.

We have changed ‘pairwise imputation’ to ‘pairwise deletion’ for clarity.

4) Neumann et al. is once referenced as 2020 and once as 2022, I believe it should be 2022 in both cases.

Thank you, we have now amended this.

5) Hair color is mentioned in the manuscript as negative control, but I could not find mentions of the results. Perhaps it would be good to add a sentence in the results, that the hair color PGS was not associated with child psychopathology (according to my understanding of the supplementary tables).

The hair colour PGS was not associated with child psychopathology. We provide a statement of this at page 12 line 8 of the results section: “Our negative control was not favoured over the null model.”